Associations of the volume and proportion of vigorous-intensity physical activity with all-cause, cardiovascular, and cancer mortality: a systematic review and meta-analysis

Wang Dechao 1
Wu Lina 2
Yan Lixue 1
Yang Huan 1
Huang Xiaoli 1
Wang Zongping 1
Guan Yanfei yanfei.guan@ynu.edu.cn 1
1 School of Physical Education, Yunnan University , Kunming , Yunnan Province , China
2 School of Cancer and Pharmaceutical Sciences, Faculty of Life Sciences and Medicine, King’s College London , London , United Kingdom
Alpuche Juan
Electronic publication date: 2025 Jun 4
Publication date: 2025
Volume: 13
Electronic Location ID: e19538
Received 2024 Dec 6; Accepted 2025 May 7
Copyright: ©2025 Wang et al.
Copyright year: 2025
Copyright holder: Wang et al.
License: This is an open access article distributed under the terms of the Creative Commons Attribution License, which permits unrestricted use, distribution, reproduction and adaptation in any medium and for any purpose provided that it is properly attributed. For attribution, the original author(s), title, publication source (PeerJ) and either DOI or URL of the article must be cited.
License URL: https://creativecommons.org/licenses/by/4.0/

Keywords: Vigorous-intensity physical activity, Mortality, CVD, Dose-response

Funding: Research Initiation Support Program for Talented Scholars at Yunnan University CY22623101 The Recommended Graduate Research Innovation Program of Yunnan University TM-23236942 This work was supported by the Research Initiation Support Program for Talented Scholars at Yunnan University (No. CY22623101) and the Recommended Graduate Research Innovation Program of Yunnan University (No. TM-23236942). The funders had no role in study design, data collection and analysis, decision to publish, or preparation of the manuscript.

==============================
Objective

This study aims to systematically investigate the associations of varying volumes of vigorous-intensity physical activity (VPA) and its proportion to moderate-to-vigorous intensity physical activity (MVPA) with all-cause, cardiovascular disease (CVD), and cancer mortality.

Methods

The review was registered in the PROSPERO (CRD42024525067). Web of Science, Embase, and PubMed were searched from inception to March 22, 2024. Empirical studies that investigated the effects of VPA compared to light-to-moderate intensity physical activities (LMPA) on all-cause, CVD, and cancer mortality were included. Additionally, studies that reported the effects of the proportion of VPA (relative to MVPA) on these mortality risks were also included.

Results

In total, 20 studies were included in the analyses. A curvilinear inverse dose-response relationship was observed between the volume of VPA and all-cause, CVD, and cancer mortality. Engaging in 180 minutes of VPA per week was associated with a substantial reduction in mortality risk: 22% for all-cause mortality, 23% for CVD mortality, and 14% for cancer mortality, compared to LMPA. Further increases in VPA volume yielded only modest additional benefits. Furthermore, a U-shaped inverse dose-response relationship was observed between the proportion of VPA (relative to MVPA) and all-cause as well as CVD mortality. Compared to 0% VPA, a 37.5% VPA (relative to MVPA) was associated with the greatest reduction in all-cause mortality (HR = 0.90, 95% CI [0.88–0.93]) and CVD mortality (HR = 0.88, 95% CI [0.83–0.94]) risk, and the size of the reduction remained stable when VPA constituted 30–60% of MVPA.

Conclusion

Engaging in more than 180 minutes of VPA per week is associated with a substantial reduction in risks of all-cause, CVD, and cancer mortality. Maintaining VPA at 30-60% of total MVPA appears to be associated with maximal reduction in all-cause and CVD mortality risks.

Introduction

The role of physical activity intensity in promoting human health has aroused great attention (Ekelund et al., 2023; Fosstveit et al., 2024). Recent observational studies utilizing accelerometers to record the amount and intensity of physical activities have demonstrated the remarkable effectiveness of vigorous-intensity physical activity (VPA) in health promotion (García-Hermoso et al., 2021). For instance, as little as 4.4 min of VPA per day was associated with a 26% to 30% reduction in all-cause mortality risk and a 24% to 32% reduction in cardiovascular disease (CVD) mortality risk, in a sample of non-exercisers from the UK Biobank (n = 25,241) (Stamatakis et al., 2022). Similarly, in another UK Biobank subsample (n = 22,398), a median of 4.5 min of daily VPA was associated with a 31% reduced risk of cancer related to lack of physical activity (Stamatakis et al., 2023). These findings highlight that even short durations of VPA can provide substantial health benefits.

While with potentially greater variability, self-reported data allows large-scale investigations that include diverse populations, offering insights not easily captured by accelerometer-based research. Overall, individuals with self-reported daily VPA at low, moderate, or high volumes demonstrated a greater reduction in all-cause and cause-specific mortality risks compared to those engaging only in light-to-moderate intensity physical activities (LMPA) (Ding et al., 2022; Wang et al., 2021; Zhao et al., 2019). However, previous studies may have overlooked critical issues. First, the volume of VPA varies across studies, whether data were recorded using accelerometers or self-reports, making it unclear what the optimal volume of VPA is for reducing mortality risks. Second, whether there is a dose–response relationship between the volume of VPA and mortality risk is unclear. To address these issues, a systematic review and meta-analysis to examine the association between the VPA volume and these mortality risks, and the potential dose–response relationship between them is needed. This would also help determine the amount of VPA required to achieve substantial mortality risk reduction and identify whether an upper limit exists for causing health benefits.

In practice, people usually engage in both VPA and moderate-intensity physical activity (MPA) during one session. The World Health Organization (WHO) physical activity guidelines recommend adults accumulate at least 150 min of MPA per week, or 75 min of VPA per week, or an equivalent combination (where 1 min of VPA is considered equivalent to 2 min of MPA), specifying the required amount of moderate-to-vigorous intensity physical activity (MVPA) (Bull et al., 2020). These three activity patterns (MPA, VPA, and MVPA) can be considered as different proportions of VPA to MVPA, corresponding to 0% VPA, 100% VPA, and any other proportions of VPA between these two extremes. By adjusting for the total volume of MVPA, previous studies reported that a higher proportion of self-reported VPA to MVPA was beneficial for reducing all-cause and cause-specific mortality risks, but this might stand only within a certain range (Lee et al., 2022; Lopez et al., 2019; Mu et al., 2022). Beyond this specific range, increasing the proportion of VPA may lead to diminishing health benefits, while controversy exists regarding this specific range (Lee et al., 2022; Lopez et al., 2019; Mu et al., 2022; Rey Lopez et al., 2020). A dose–response meta-analysis of current large-scale observational studies is essential to further examine the relationship between the proportion of VPA (relative to MVPA) and these mortality risks, as well as to determine the optimal VPA proportion associated with the lowest mortality risks.

Therefore, the purpose of this systematic review and meta-analysis is twofold: (1) to assess the association between the volume of VPA and the risks of all-cause, CVD, and cancer mortality; (2) to explore the dose–response relationship between the proportion of VPA (relative to MVPA) and the risks of all-cause, CVD, and cancer mortality.

Methods

We registered the study protocol with PROSPERO (CRD42024525067), and the reporting adhered to the Preferred Reporting Items for Systematic Reviews and Meta-Analysis (PRISMA) (Liberati et al., 2009). Given that the included studies were observational and based on large sample sizes or nationally representative populations, we also followed the Meta-analysis of Observational Studies in Epidemiology (MOOSE) (Stroup et al., 2000) guidelines.

Search strategy and selection criteria

One author (DW) conducted systematic literature searches in Web of Science, Embase, and PubMed from database inception to March 22, 2024. The search strategy for each database is shown in Table S1. Four authors (DW, LY, HY, and XH) screened and selected studies for inclusion in this review. Any disagreements during the review process were resolved through consensus between the two reviewers (DW and YG). Only studies written in English were included, and references from the selected studies and reviews were manually checked. Conference abstracts and unpublished studies were excluded.

Studies were eligible if they met all of the following criteria: (1) had a prospective cohort, repeated cross-sectional, or nested case-control study design, with a large sample size or based on nationally representative populations; (2) had a follow-up period of at least 3 years; (3) the outcomes were all-cause, CVD, or cancer mortality. Additionally, to achieve the two objectives of this research, we also required that studies met the following criteria: (1) The intensity categories of physical activity (i.e., light-, moderate-, and vigorous-intensity physical activity) in which participants engaged, including a comparison between VPA and LMPA, were reported. VPA was defined as activities with ≥ 6 metabolic equivalent tasks (METs) or those causing heavy sweating or significant increases in breathing or heart rate. (2) For studies reporting the proportion of VPA to MVPA, information on the number of participants/person-years and cases for each level of proportion of VPA to MVPA should be provided. (3) Studies focusing on cohorts with specific diseases (e.g., hypertension, diabetes, or cancer) or a specific sport (e.g., marathon, running, or cycling) were excluded. Randomized controlled trials (RCTs) were deemed ineligible, as they are generally unfeasible for large-scale studies requiring long-term follow-up, especially when assessing mortality outcomes and the comparative effects of VPA and LMPA.

Data extraction and risk of bias assessment

Two authors (DW and LY) extracted the following details from the included publications: first author, publication year, study time and location, cohort characteristics, follow-up duration, physical activity measurement methods, covariates in the adjusted model, mortality outcomes, and participant numbers, age, sex, and inclusion criteria. Hazard ratios (HRs) and corresponding 95% confidence intervals (CIs) were extracted by one author (DW) and independently reviewed by another author (LY) to ensure the selection of the most appropriately adjusted estimates.

Two authors (DW and LY) independently assessed the quality of the included studies using a modified version of the Newcastle-Ottawa Scale (Peterson et al., 2011), with a maximum possible score of eight stars (Table S2). Discrepancies were resolved through discussion with a third reviewer (YG). To complement this, we also used the ROBINS-I tool (Sterne et al., 2016) to evaluate the risk of bias, defining the target population as healthy adults, with VPA as the intervention and LMPA as the comparator. The outcomes of interest included all-cause, CVD, and cancer mortality. Major confounders (e.g., age, sex, smoking status, alcohol consumption, dietary factors or BMI, individual-level socioeconomic status, physical activity intensity, and family history of CVD) were considered. ROBINS-I results are presented in Fig. S1.

Data synthesis and statistical analysis

Statistical analyses were performed using Stata statistical software (version 14) (Orsini et al., 2012). The random-effects model by DerSimonian & Laird (2015) was used to pool the HRs, accounting for variability in true effects across studies, particularly arising from differences in assessment methods. A restricted cubic spline dose–response meta-analysis was conducted to evaluate the association between the minutes of VPA per week (volume) and the risks of all-cause, CVD, and cancer mortality. This method was employed because of its flexibility in modeling non-linear relationships (Orsini et al., 2012), and has been widely validated and applied in previous studies. To pool study-specific HRs, we applied the method originally proposed by previous studies (Greenland & Longnecker, 1992; Orsini et al., 2012) and subsequently extended for use with time-to-event data (Blond et al., 2020). Due to limited data in the upper tail of the exposure distribution (400–750 minutes/week), numerical instability occurred when estimating the full variance–covariance matrix required for the random-effects model. Given these computational challenges, we opted for a fixed-effects model (Orsini et al., 2012). The included studies exhibited a high level of methodological consistency, including study design, population scale, physical activity assessment, and outcome measures, supporting the appropriateness of the fixed-effects approach in this context. Additionally, the knot locations for the restricted cubic spline regression were set at the 25th, 50th, and 75th percentiles of the overall exposure distribution (Gauthier, Wu & Gooley, 2020; Orsini et al., 2012). Non-linearity was assessed by testing whether the coefficient of the second spline transformation was equal to zero and by performing a likelihood ratio test. For this analysis, we required the number of cases, person-years, the quantification of the varying volumes of VPA exposure, and HRs with corresponding 95% CIs of at least three categories. If the original article did not provide person-year data, the distributions of person-years were estimated based on the total number of cases, participants, and follow-up period, using an established method from the previous study (Aune et al., 2012). The quantification of VPA volume was calculated using the midpoint between the lower and upper limit ranges. For open-ended categories, the width was assumed the same as the adjacent categories. The original exposure categories and our calculations of minutes of VPA were presented in Table S3. In addition, studies lacking essential data (e.g., person-years) were excluded from the dose–response meta-analysis but were included in the random-effects model (DerSimonian & Laird, 2015) to estimate the summary effects of the highest volume (when the VPA volume was categorized into different levels) of VPA vs. LMPA (no VPA) on all-cause, CVD, and cancer mortality risks. Sensitivity analyses were performed by removing one study at a time. Additionally, studies identified as potential sources of heterogeneity were included in subgroup analyses.

Furthermore, the non-linear dose–response association between the proportion of VPA (relative to MVPA) and the risks of all-cause mortality as well as CVD mortality was assessed by conducting another restricted cubic spline dose–response meta-analysis, employing a random-effects model (Greenland & Longnecker, 1992; Orsini et al., 2012). Knot locations for the restricted cubic spline regression were set at the 10th, 50th, and 90th percentiles of the overall exposure distribution (Harrell, 2001; Orsini et al., 2012). In this analysis, the proportion of VPA to MVPA was considered as the “dose” variable. If the proportion of VPA to MVPA in the included studies was reported as ranges, the dose was estimated using the midpoints between the lower and upper limits of these ranges. The proportion of VPA to MVPA was primarily calculated in two steps: first, total MVPA was estimated by doubling the duration of VPA (since 1 min of VPA is considered equivalent to 2 min of MPA) and adding the duration of MPA. The proportion of VPA was then derived by dividing the duration of VPA by total MVPA. Details of the ascertainment and calculation of the proportion of VPA to MVPA for the included studies are available in Tables S4 and S5. In addition, we also estimated summary HRs for the highest vs. lowest proportions of VPA (relative to MVPA) for all-cause and CVD mortality risks using a random-effects model.

Since a maximum of eight studies were included in the two dose–response meta-analyses, sub-group analyses were not performed. The decorrelated residuals versus exposure plots were visually inspected to assess the model fit and determine if the fit depended on the exposure level (post hoc) (Discacciati, Crippa & Orsini, 2017). Publication bias was assessed by visually inspecting funnel plots of the HR (estimated with per unit increase in the exposure as a linear predictor) against the SE of the log (HR) and by conducting Egger’s regression test for funnel plot asymmetry (Egger et al., 1997; Sterne & Harbord, 2004).

Heterogeneity was assessed using the inconsistency index (I2) (Borenstein et al., 2017) with values of 25%, 50%, and 75% corresponding to low, moderate, and high heterogeneity, respectively. All tests were two-sided, with 0.05 to be the significance threshold.

Results

Study screening

After starting with 1,922 studies, 694 were excluded due to duplication, and 886 were excluded based on title and abstract screening. Full texts were retrieved for 342 studies with the potential for inclusion. Twenty studies were finally included in the analysis (Arem et al., 2015; Ding et al., 2022; Duarte Jr et al., 2024; Evenson, Wen & Herring, 2016; Gebel et al., 2015; Kikuchi et al., 2018; Kushi et al., 1997; Lahti et al., 2014; Lee et al., 2022; Lee, Hsieh & Paffenbarger, 1995; Lee & Paffenbarger, 2000; Leitzmann et al., 2007; Lopez et al., 2019; Mu et al., 2022; Rockhill et al., 2001; Shiroma et al., 2014; Wang et al., 2021; Wen et al., 2011; Yu et al., 2003; Zhao et al., 2019). A PRISMA flow chart presenting this progress is shown in Fig. 1 (Liberati et al., 2009).

Figure 1 PRISMA flowchart of the literature search and literature inclusion.

Study characteristics

Characteristics of the included studies were shown in Table S6. Of the 20 studies meeting the inclusion criteria, data from eight, seven, and six studies were used to estimate the dose–response relationship between the volume of VPA per week and HRs for all-cause, CVD, and cancer mortality, respectively. Additionally, HRs associated with the highest volume of VPA compared to LMPA (no VPA) were pooled from studies on all-cause mortality (n = 15), CVD mortality (n = 10), and cancer mortality (n = 7). For the analysis of the dose–response relationship between the proportion of VPA (relative to MVPA) and all-cause mortality, as well as pooled HRs for the highest versus the lowest proportion, nine cohorts from seven studies (Gebel et al., 2015; Kikuchi et al., 2018; Lee et al., 2022; Lopez et al., 2019; Mu et al., 2022; Shiroma et al., 2014; Wang et al., 2021) were included. For CVD mortality, the analysis included six cohorts from five studies (Lee et al., 2022; Lopez et al., 2019; Mu et al., 2022; Shiroma et al., 2014; Wang et al., 2021). In all of these seven studies (on all-cause and CVD mortalities), six calculated the proportion of VPA to MVPA based on the assumption that 1 min of VPA was equivalent to 2 min of MPA. Only in one study (Shiroma et al., 2014), the proportion of VPA to MVPA was calculated by directly dividing VPA by MVPA energy expenditure in metabolic equivalents (METs). All included studies had an average range of participants’ age over 40, except for one study (Wen et al., 2011) which included participants starting at 20 years old. All studies had a follow-up period of more than five years and utilized self-reported data of physical activity. The included studies were conducted in nine different countries, with 11 studies originating from the United States.

Risk of bias assessment

Using the Newcastle-Ottawa Scale, one study received the highest quality assessment rating of eight stars, while nine studies received seven stars. Additionally, all studies received at least five stars (Table S2). Some studies did not receive a potential star in the “Comparability of cohorts on the basis of the design or analysis” item due to not meeting all criteria for variable adjustments.

In terms of risk of bias assessment using the ROBINS-I tool, evaluations were conducted across seven domains (Fig. S1). Regarding confounding, fourteen studies were classified as having a serious risk of bias, while six were assessed as having a moderate risk. For selection of participants, all studies were rated with a moderate risk of bias. Conversely, in the domains of classification of exposures, deviations from intended exposures, and measurement of outcomes, all studies were assessed with a low risk of bias. In the domain of missing data, four studies were rated as having no information, while another four were considered to have a serious risk of bias. For the selection of reported results domain, thirteen studies were judged to have a low risk, and seven were rated as having a moderate risk. Overall, all of included studies had moderate or serious risk of bias, while none had a critical risk of bias, supporting the usefulness of the included evidence, although findings should be interpreted with caution.

Meta-analysis

The volume of VPA and all-cause, CVD, and cancer mortality risks

Dose–response analyses.

A curvilinear inverse relationship was shown between the volume of VPA and the risks of all-cause, CVD, and cancer mortality (P for non-linearity < 0.001) (Fig. 2A, Table S7). With 0 min of VPA per week as the reference, any amount of VPA was associated with a lower risk of all-cause, CVD, and cancer mortality, with substantial mortality risk reduction occurring at 180 min per week (all-cause mortality HR = 0.78, 95% CI [0.76–0.79], I2 = 88.46%; CVD mortality HR = 0.77, 95% CI [0.73–0.80], I2 = 70.13%; cancer mortality HR = 0.86, 95% CI [0.82–0.90], I2 = 46.64%). Further increase in the volume of VPA beyond 180 min per week yielded only modest additional benefits, and reduced benefits were observed when VPA volume exceeded 450 min per week in the dose–response curves for all-cause and CVD mortality risk. Funnel plots for the HRs related to all-cause, CVD, and cancer mortality risks, plotted against the standard error of the log (HR), were asymmetric (Fig. S2). Egger’s test yielded P-values of 0.037, 0.063, and 0.094 for these risks, indicating potential publication bias (Lau et al., 2006; Terrin et al., 2003). The decorrelated residuals versus exposure plots showed a less adequate fit at >400 min of VPA per week for all-cause and cancer mortality risk models (Fig. S3).

Figure 2 (A) Dose–response relationship between the volume of VPA and all-cause, CVD, and cancer mortality. (B) Forest plot summary of the association between the highest volume of VPA vs. LMPA and all-cause, CVD, and cancer mortality.

The highest volume of VPA vs. LMPA (no VPA).

Eight studies were included to compare the effects of the highest volume (the VPA volume was categorized into different levels) of VPA versus LMPA on mortality risks (Fig. 2B). Engaging in the highest volume of VPA was associated with reduced risks of all-cause (HR = 0.75, 95% CI [0.70–0.80], I2 = 77.3%), CVD (HR = 0.71, 95% CI [0.63–0.81], I2 = 64.8%), and cancer mortality (HR = 0.85, 95% CI [0.79–0.92], I2 = 46.8%) compared to LMPA (Fig. 2B). Even though Egger’s test yielded P-values of 0.119, 0.082, and 0.221, the asymmetry observed in the funnel plot (Figs. S4–S6) suggests that publication bias could not be conclusively ruled out (Lau et al., 2006; Terrin et al., 2003). Sensitivity analyses were performed by removing one study at a time (Figs. S7–S9). Subgroup analyses were conducted post hoc based on sensitivity analyses and variations in study-level characteristics identified as potential contributors to heterogeneity, such as methodological differences and population variations. The subgroup analyses reduced heterogeneity and yielded consistent results with the overall study findings, with HR of 0.76 (95% CI [0.73–0.80], I2 = 26.3%) for all-cause mortality risk, 0.73 (95% CI [0.68–0.79], I2 = 0%) for CVD mortality risk, and 0.88 (95% CI [0.83–0.93], I2 = 10.3%) for cancer mortality risk, respectively (Figs. S10–S12).

The proportion of VPA (relative to MVPA) and all-cause and CVD mortality risks

Dose–response analyses.

A U-shaped inverse relationship between the proportion of VPA (relative to MVPA) and all-cause as well as CVD mortality risk (P for non-linearity < 0.001) was observed (Fig. 3A and Table S8). Using 0% of VPA to MVPA as the reference level, the lowest HRs were observed when VPA comprised 37.5% of MVPA for all-cause (HR = 0.90, 95% CI [0.88–0.93]) and CVD mortality risk (HR = 0.88, 95% CI [0.83–0.94]), with only minor variations in HRs within 30–60% of VPA to MVPA. However, a further higher proportion of VPA beyond 60% was associated with diminished effectiveness in reducing mortality risk; and at the right end of the shapes, engaging in VPA alone (100% VPA to MVPA) showed significantly reduced benefits for all-cause (HR = 0.95, 95% CI [0.90–1.01]) and CVD mortality risks (HR = 0.97, 95% CI [0.87–1.08]). In contrast, the optimal combination of VPA and MPA, which was 37.5% of VPA to MVPA, produced better outcomes with a 10% reduction in all-cause and a 12% reduction in CVD mortality risk compared to MPA alone, and with a 5% reduction in all-cause and a 9% reduction in CVD mortality risk compared to VPA alone.

There was a relatively high between-study variance (I2 = 82.4% for all-cause mortality risk and I2 = 70.7% for CVD mortality risk) in this analysis. The funnel plot was symmetric, with P-values of 0.512 and 0.499 from Egger’s test for all-cause and CVD mortality risk, respectively (Fig. S13). The decorrelated residuals versus the proportion of VPA to MVPA plot showed a good fit between the residuals and exposure for the all-cause and CVD mortality risk models (Fig. S14). Additionally, two studies (Lopez et al., 2019; Wang et al., 2021) that assessed the proportion of VPA (relative to MVPA) and cancer mortality risk found that the risk initially decreased but subsequently increased as the proportion rose, suggesting that the optimal benefit was confined to a specific range.

Figure 3 (A) Dose–response relationship between the proportion of VPA (to MVPA) and the all-cause and CVD mortality risk. (B) Forest plot summary of the association between highest vs. lowest proportions of VPA (to MVPA) and all-cause and CVD mortality risk.

The highest proportion of VPA to MVPA vs. the lowest.

We calculated the HRs for the highest versus lowest proportion of VPA (to MVPA) for all-cause and CVD mortality (Fig. 3B). Engaging in the highest proportion of VPA was associated with a reduced risk of all-cause (HR = 0.93, 95% CI [0.85–1.01], I2 = 88.3%) and CVD mortality risk (HR = 0.90, 95% CI [0.79–1.02], I2 = 64.8%). However, a substantial between-study variance was observed, with the midpoint of the range of the highest proportion of VPA (relative to MVPA) varying significantly across original studies (ranging from 0.65 to 1), which might have impacted the statistical power of the analysis. Funnel plots for both all-cause and CVD mortality risk suggest no publication bias (Fig. S15), further supported by Egger’s test (P = 0.086 and 0.083, respectively).

Discussion

This study is the first to investigate the association of VPA volume and proportion (relative to MVPA) with health outcomes by comprehensively analyzing results from large-scale studies with long-term follow-up. For the first purpose, we systematically evaluated the relationship between the volume of VPA and the risks of all-cause, CVD, and cancer mortality, and compared the effects of the highest volume (when the VPA volume was categorized into different levels) of VPA versus LMPA (no VPA). The results revealed a non-linear relationship between the volume of VPA and the mortality risks, with even a small amount of VPA associated with significant mortality risk reduction. Notably, substantial benefits were observed at 180 min of VPA per week, associated with a 22%, 23%, and 14% reduction in all-cause, CVD, and cancer mortality risk, respectively. A notable decline in benefits was observed when VPA volume exceeded 450 min per week in the dose–response curves for all-cause and CVD mortality risk. However, limited data on VPA exposure beyond 400 min per week may weaken the statistical power of this finding. Moreover, the reductions in all-cause and CVD mortality observed at the highest VPA volumes vs. LMPA (no VPA) were greater than those estimated from the dose–response analysis (180 min of VPA per week), with reductions of 25% vs. 22% for all-cause mortality, and 29% vs. 23% for CVD mortality. For these reasons, we can’t rule out the effectiveness of VPA volume exceeding 450 min in reducing the mortality risks. However, it is clear that beyond 180 min of VPA per week, the additional benefits are modest, accompanied by increasing uncertainty, as reflected by the wide confidence intervals. These findings suggest that individuals can incorporate 180 min or more of VPA per week to efficiently minimize the risk of all-cause, CVD, and cancer mortality.

A couple of studies have reported supportive results to our findings. A large-scale study involving 136,766 participants across 21 countries found that, at each level of the total volume of physical activity, participants engaging in VPA exhibited lower CVD mortality risk compared to those without engaging in VPA (Lear et al., 2023). Another study found that approximately 30–35 min of VPA per week and 500 min of MVPA per week provided comparable benefits in mitigating the association between abdominal obesity and incident cardiovascular disease (Sanchez-Lastra et al., 2024). Together with the findings of the present study, the crucial role of engaging in VPA for efficient health promotion is highlighted, which extends beyond merely considering the volume of physical activity. In addition, results from the original studies have generally indicated that the greatest reduction in all-cause and CVD mortality risk was achieved when participants engaged in 150–300 min of MPA and at least 150 min of VPA per week (Lee et al., 2022; Mu et al., 2022; Rockhill et al., 2001; Wang et al., 2021), which aligns with our results demonstrating a substantial reduction in mortality risks associated with VPA volume exceeding 180 min per week. Collectively, these findings underscore the importance of VPA and its volume in maximizing health benefits.

Another important finding of the present study is the association between the proportion of VPA (relative to MVPA) and the mortality risks. WHO physical activity guidelines propose three intensity patterns (MPA, VPA, and MVPA), with 1 min of VPA equivalent to 2 min of MPA for daily accumulation of physical activity (Bull et al., 2020). Based on this principle, we systematically reviewed studies examining the association between the proportion of VPA (relative to MVPA) and mortality risks, and observed a U-shaped inverse relationship between the proportion of VPA (relative to MVPA) and all-cause and CVD mortality risk. Our findings also demonstrate that engaging in MVPA, with VPA comprising 30–60% (of the total MVPA), is more effective in reducing all-cause and CVD mortality risks compared with MPA alone (0% VPA to MVPA) or VPA alone (100% VPA to MVPA).

Considering the fact that achieving a 100% VPA proportion within the total MVPA was usually not common, we summarized HRs for the highest VPA proportions (not necessarily 100%) versus the lowest (MPA only). The summary HRs for all-cause mortality (HR = 0.93, 95% CI [0.85–1.01]) and CVD mortality (HR = 0.90, 95% CI [0.79–1.02]) showed wide confidence intervals, indicating uncertainty regarding their protective effects. Furthermore, in the original studies, although the highest VPA proportions ranging from 65% to 100% showed significant variability in reducing mortality risks, they were generally associated with reduced benefits compared to the optimal VPA proportion in each study. Together with the results in the present study, incorporating 30–60% VPA in MVPA appears optimal for reducing all-cause and CVD mortality risks, while a further increase in the proportion of VPA shows reduced benefits. This finding suggests that practitioners advocate a balanced approach to physical activity, incorporating 30–60% VPA in daily MVPA to efficiently lower the risks of all-cause and CVD mortality.

The 2020 WHO guidelines recommend that adults engage in at least 150–300 min MPA, or at least 75–150 min of VPA, or an equivalent combination of the two per week for substantial health benefits (Bull et al., 2020). The guidelines also note that exceeding 300 min MPA, or 150 min of VPA, or the equivalent combination, can provide increasing general health benefits (Bull et al., 2020). Our findings that 180 min of VPA per week for individuals engaging in only LMPA is associated with substantial reductions in all-cause, CVD, and cancer mortality risks (with only modest additional benefits observed beyond this threshold), are consistent with the guidelines, and offer a valuable supplement by addressing specific mortality outcomes. Moreover, although the WHO guidelines allow substitution between MPA and VPA for general health benefits, our findings emphasize that having VPA contribute 30–60% of total MVPA is associated with maximal reductions in all-cause and CVD mortality risks, underscoring the particular importance of VPA.

A potential issue of the findings is about the effects of VPA proportion (relative to MVPA) across different levels of MVPA volumes. Lee et al. (2022) analyzed the impacts of VPA within stratified amounts of total MVPA on all-cause and CVD mortality risks and found that increasing the VPA proportion was associated with lower mortality risk when MVPA was less than 300 min per week, but showed diminished benefits when MVPA exceeded 300 min. Supportively, a recent study (Strain et al., 2020) involving 96,476 participants demonstrated that a higher proportion of energy expenditure from MVPA in total physical activity was associated with lower all-cause mortality risk, particularly at total physical activity with lower levels of energy expenditure, while the benefits diminished at higher levels of energy expenditure. Together, these findings imply that the benefits of increasing VPA volume may be underrecognized at lower volumes of physical activity, but may be overestimated at higher volumes. Therefore, our findings regarding the 30–60% VPA proportion may be more reliable in the context of lower total volumes of physical activity. Based on our first conclusion that 180 min of VPA per week (for individuals engaging only in LMPA) yields substantial benefits, the potential gain from maintaining this proportion by increasing LMPA remains uncertain. Further research is needed to explore the association between VPA proportions and mortality risks across varying levels of physical activity volume.

The strength of this study lies in the quantification of the association of varying volumes and proportions of VPA (relative to MVPA) with the risks of CVD, cancer, and all-cause mortality through non-linear modeling. A common limitation in previous studies examining the effects of physical activity intensity on health outcomes is the failure to account for the volume of physical activity, which would diminish the credibility of the results. The present research addresses this limitation by considering the proportion of VPA (relative to MVPA) as a dose, allowing analysis of the effects of VPA versus MPA as well as the joint effects of VPA and MPA on mortality risks, thereby enabling more robust and reliable conclusions. Publication bias was evaluated by funnel plots and Egger’s regression test in each analysis to ensure the robustness of the findings.

There are several limitations in this study. Firstly, all included studies relied on self-reported physical activity data, which introduces potential measurement bias. A potential limitation of self-reported data is the tendency to overestimate the volume of VPA, and individual variability in reporting, which may result in an underestimation of the relationship between VPA and health outcomes (Prince et al., 2008; Ramakrishnan et al., 2021). However, the self-reported measurement allows for large-sample research, which is important for investigating the association with the mortalities. Moreover, a fixed-effects model was used to analyze the association between the volume of VPA and the outcomes. While this approach allowed for stable estimation, it may lead to underestimation of uncertainty in the pooled estimates, and this limitation should be considered when interpreting and applying the findings. Furthermore, since the studies included in our research predominantly used statistical models that adjusted for total MVPA volume, few have directly examined the relationship between VPA proportion and mortality risk across different levels of MVPA volume. In addition, considerable between-study heterogeneity was observed in certain analyses, while attempts to identify the sources of heterogeneity through subgroup analyses were limited by the small number of studies, which may reduce the statistical power.

Conclusion

This meta-analysis demonstrated a curvilinear inverse dose–response relationship between the volume of VPA and the risks of all-cause, CVD, and cancer mortality. Engaging in at least 180 min of VPA per week was associated with substantial reductions in these mortality risks. Additionally, a VPA proportion of 30–60% of total MVPA appears to be associated with maximal reductions in all-cause and CVD mortality risks, as indicated by a U-shaped inverse dose–response relationship between the proportion of VPA (relative to MVPA) and all-cause as well as CVD mortality risk. Overall, our findings address the importance of the VPA volume and its proportion (relative to MVPA) in health promotion, suggest incorporating a certain amount of VPA (>180 min per week) into daily physical activities, and specify an optimal combination of VPA and MPA (30–60% VPA to MVPA) in reducing all-cause, CVD, and cancer mortality risks. Future research is needed to further investigate the effects of VPA proportion on mortality risks by analyzing these effects across different levels of MVPA volume.

Supplemental Information

Supplemental Information 1 PRISMA checklist

Supplemental Information 2 Rationale

Supplemental Information 3 Supplementary figures and tables

These data were derived from the studies included in this analysis.

Additional Information and Declarations

Competing Interests

Author Contributions

Data Availability

The authors declare there are no competing interests.

Dechao Wang conceived and designed the experiments, performed the experiments, analyzed the data, prepared figures and/or tables, authored or reviewed drafts of the article, and approved the final draft.

Lina Wu performed the experiments, analyzed the data, authored or reviewed drafts of the article, and approved the final draft.

Lixue Yan performed the experiments, analyzed the data, authored or reviewed drafts of the article, and approved the final draft.

Huan Yang performed the experiments, analyzed the data, authored or reviewed drafts of the article, and approved the final draft.

Xiaoli Huang performed the experiments, authored or reviewed drafts of the article, and approved the final draft.

Zongping Wang conceived and designed the experiments, authored or reviewed drafts of the article, and approved the final draft.

Yanfei Guan conceived and designed the experiments, performed the experiments, authored or reviewed drafts of the article, and approved the final draft.

The following information was supplied regarding data availability:

This is a systematic review/meta-analysis. All data were obtained from original studies and are available in the Supplemental Information.

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
