# Peer review of "Associations of the volume and proportion of vigorous-intensity physical activity with all-cause, cardiovascular, and cancer mortality: a systematic review and meta-analysis"

_PeerJ, doi:10.7717/peerj.19538_

## Round 0.1 · original submission · Minor Revisions

Our reviewers' comments call for clearer English usage, alignment with WHO guidelines, and clarification of the 180-minute VPA threshold versus the 30-60% MVPA ratio. They question the methodological rigour (exclusion of RCTs, limited assessment of bias), emphasise that the findings are associative rather than causal, and recommend refining the analyses for greater statistical consistency, interpretability and practical application.

**Language Note:** The review process has identified that the English language must be improved. PeerJ can provide language editing services - please contact us at [email protected] for pricing (be sure to provide your manuscript number and title). Alternatively, you should make your own arrangements to improve the language quality and provide details in your response letter. – PeerJ Staff

Reviewer 1 ·

Basic reporting

The manuscript would benefit from additional editing for clarity and consistency, particularly in ensuring appropriate English language usage throughout. Some sentences are complex and may be difficult for readers to interpret. Consider refining the text to improve readability, eliminate redundancy, and enhance precision in describing findings.

Experimental design

No comment.

Validity of the findings

The study presents compelling evidence that engaging in ≥180 minutes of vigorous-intensity physical activity (VPA) per week is associated with maximal reductions in all-cause, cardiovascular, and cancer mortality. Additionally, the findings suggest that VPA should ideally constitute 30-60% of total moderate-to-vigorous physical activity (MVPA) for optimal benefits. However, there are several areas where further clarification would enhance the manuscript’s practical applicability and alignment with public health recommendations.

1. Alignment with WHO Guidelines
The World Health Organization (WHO) currently recommends:

At least 75 minutes of VPA per week or 150 minutes of moderate-intensity physical activity (MPA) per week for substantial health benefits.
For additional health benefits, adults should increase their moderate-intensity activity to 300 minutes per week or engage in an equivalent combination.
Muscle-strengthening activities should be performed at least twice per week.
In contrast, this study suggests that 180 minutes of VPA per week provides optimal health benefits. However, the manuscript does not explicitly address how this recommendation aligns with or differs from WHO’s existing thresholds.

Does the dose-response analysis indicate that WHO’s current recommendation of 75 minutes of VPA per week is insufficient for maximal health benefits?
Should public health recommendations consider revising the current VPA guidelines based on these findings?
The WHO also allows for an equivalent combination of VPA and MPA. How does this study’s 30-60% ratio recommendation fit within WHO’s framework for equivalent activity?
Clarifying this would enhance the study’s public health relevance and ensure that its findings are interpreted in the context of existing global recommendations.

2. Clarification of the Relationship Between the 180-Minute Threshold and the 30-60% Ratio
It is unclear whether the 180-minute threshold represents an absolute target for VPA alone or whether it should be interpreted within the context of a larger MVPA recommendation where VPA constitutes only 30-60% of total MVPA.
If 180 minutes of VPA represents only 30-60% of total MVPA, then total weekly MVPA would need to be much higher (300-600 minutes per week)—a level significantly exceeding WHO recommendations.
Conversely, if 180 minutes of VPA is an independent goal, then the 30-60% ratio would apply only in cases where individuals engage in both moderate and vigorous activity but would not require additional MPA for those meeting the 180-minute VPA threshold.
3. Practical Implications for Public Health Guidance
If the study’s recommendations necessitate 180 minutes of VPA plus additional MPA to maintain the 30-60% ratio, this would significantly exceed WHO’s current guidance.
If, instead, 180 minutes of VPA is sufficient as a standalone recommendation, then public health messaging should clarify that additional MPA is not necessarily required for those reaching this threshold.
The authors may wish to explicitly state whether their findings suggest a revision to current physical activity guidelines or whether their results simply refine recommendations for optimal, rather than minimum, health benefits.

·

Basic reporting

The explanations for grammatical refinements (Abstract and Discussion) should be formatted more uniformly. Instead of listing them separately with numbers, consider integrating them into a paragraph or bullet points for smoother reading.

Example:
"Engaging in 180 minutes of VPA per week was associated with a substantial reduction in mortality risk: 22% for all-cause mortality, 23% for CVD mortality, and 14% for cancer mortality, compared to LMPA."
(Avoids repetition of "reduction" and improves readability by restructuring the sentence.)

"Furthermore, a U-shaped inverse dose-response relationship was observed between the proportion of VPA (relative to MVPA) and all-cause as well as CVD mortality."
(Removes redundancy—"curvilinear" is implied by "U-shaped.")

"A notable decline in benefits was observed when VPA volume exceeded 450 minutes per week in the dose-response curves for all-cause and CVD mortality risk."
(Improves clarity and readability.)

"However, limited data on VPA exposure beyond 400 minutes per week may weaken the statistical power of this finding."
(Concise while retaining the original meaning.)

"This finding suggests that practitioners advocate a balanced approach to physical activity, incorporating 30–60% VPA in daily MVPA to effectively lower the risks of all-cause and CVD mortality."
(More natural phrasing and appropriate word choice.)

Experimental design

The manuscript has significant methodological concerns that limit its reliability.
First, the study does not include RCTs, and the justification for this exclusion is insufficient, making it difficult to control for confounding factors.
Second, the risk-of-bias assessment relies solely on the Newcastle-Ottawa Scale (NOS), which is not detailed enough for observational studies; a more robust approach, such as ROBINS-I, should be considered.
Lastly, the conclusions should explicitly state that the findings indicate associations rather than causal relationships.

Validity of the findings

The validity of the findings is questionable due to the study’s reliance on observational data without sufficient control for confounding factors.
The exclusion of RCTs is not adequately justified, making it difficult to establish causality.
Additionally, the risk-of-bias assessment lacks depth, as it relies solely on the NOS rather than a more comprehensive approach like ROBINS-I.
Given these methodological limitations, the conclusions should be carefully interpreted and explicitly state that the findings indicate associations rather than causality.

Additional comments

Due to these methodological limitations, the manuscript is not suitable for publication in its current form.
This study requires substantial methodological revisions and extensive modifications to the overall manuscript.
Given the scope of these necessary changes, it is unlikely that all issues can be addressed within the current submission.
Therefore, I recommend rejecting the manuscript.

Reviewer 3 ·

Basic reporting

In general, this manuscript is well-written, with a clear structure and no significant typographical or grammatical errors, which enhances readability. The figures and tables are well-designed and effectively complement the text. Here are some minor comments about reporting:
• Please check line 73-74 for potential grammatical issue: “… of MPA) indicating a specific…”
• Please clarify the meaning of “Dose”: the proportion of VPA (to MVPA) in the method section
• In line 188, should “In six of these studies” be “in five of these studies” because it looks that there are six cohorts from five studies?
• Line 259, in this section, add the figure number for reference (figure 3)

Experimental design

Statistical analysis:
• Line 132-133: reference 20 (Greenland and Longnecker) introduced the method for odds ratio, but in your study the outcome variable is time to event variable with HR as the summary measure, clarify technical details regarding how the method in reference 20 was applied.
• In line 154, please clarify why random effect model was used for modeling association between proportion of VPA and outcome, but fixed effect model was used for modeling association between VPA volume and outcome

Validity of the findings

The results were presented for the association between volume of VPA/proportion of VPA (to MVPA) versus mortality risks based on restricted cubic spline dose-response model with fixed effects or random effects. Has the model fitting been checked and how is the model compared to other types of splines?

---

## Round 0.2 · Minor Revisions

Thanks for your corrections. Now your works is almost ready to be accepted.
Please, make minor revisions changes commented by reviewers.

Reviewer 1 ·

Basic reporting

No comment.

Experimental design

No comment.

Validity of the findings

Line 249 – 252 state, “Overall, all of included studies had moderate or serious risk of bias, while none had a critical risk of bias, supporting the usefulness of the included evidence, although findings should be interpreted with caution.”
However, in lines 290 – 285, critical risk of bias studies were included in a subgroup analysis.
“Studies identified as having a critical risk of bias in the meta-analysis were included in subgroup analyses, which reduced heterogeneity and yielded consistent results with the overall study findings, with HR of 0.76 (95% CI: 0.73 to 0.80, I² = 26.3%) for all-cause mortality risk, 0.73 (95% CI: 0.68 to 0.79, I² = 0%) for CVD mortality risk, and 0.88 (95% CI: 0.83 to 0.93, I² = 10.3%) for cancer mortality risk, respectively (Supplementary eFigure 9-11).”
Please clarify.

Additional comments

The manuscript is much improved. I’d like to thank the authors for taking the time and care to revise the document to its current form.

·

Basic reporting

I confirmed.

Experimental design

The study follows a rigorous and methodologically appropriate systematic review and meta-analysis framework. The rationale for excluding RCTs is now more clearly justified in the context of long-term mortality outcomes and the feasibility constraints of large-scale trials in this area. The inclusion criteria, search strategy, and risk-of-bias assessments (including the addition of ROBINS-I) are well described and enhance the reliability of the findings.

A strength of the revision is the explicit clarification of why fixed-effects models were used in one analysis and random-effects in another, which is now supported by computational and conceptual reasoning.

Validity of the findings

I confirmed.

Additional comments

Thank you very much for taking the time to respond and address my comments, despite the considerable effort required for the additional risk of bias checks. I truly appreciate your work.

I apologize for the additional trouble, but I would like to confirm three minor points regarding the Methods section:

1. The use of 'MOOSE' at line 127 is appropriate as long as the included studies are based on large sample sizes or nationally representative populations, and this condition is stated either before or after the sentence referencing 'MOOSE'.
This clarification may have deleted the redundant (L.145,146, L.156-158).

2. Please consider reordering the Appendix figures and tables to follow the standard sequence referenced in the manuscript and following PRISMA guidelines.

3. The paragraph (Lines 166–180) describing both NOS and ROBINS-I assessments is informative but slightly redundant. For improved clarity and readability, you may consider condensing the text by integrating overlapping descriptions and focusing on the key differences between the two tools.

Revised example
Two authors (DW and LY) independently assessed the quality of the included studies using a modified version of the Newcastle-Ottawa Scale (Peterson et al., 2011), with a maximum score of 8 stars (see Supplementary eTable 3). Discrepancies were resolved through discussion with a third reviewer (YG). To complement this, we also used the ROBINS-I tool (Sterne et al., 2016) to evaluate the risk of bias, defining healthy adults as the target population, with VPA as the intervention and LMPA as the comparator. Outcomes of interest included all-cause, cardiovascular, and cancer mortality. Major confounders (e.g., age, sex, smoking, BMI, socioeconomic status, and physical activity intensity) were considered. ROBINS-I results are presented in Supplementary eFigure 4.

4. Please recheck your PRISMA checklist 2020, especially the Location where item is reported, after you have finished rewriting all.

Reviewer 3 ·

Basic reporting

The authors have addressed all comments.

Experimental design

The authors have addressed all comments.

Validity of the findings

The authors have addressed all comments.

---

## Round 0.3 · accepted · Accept

Thank you for the improvements you have made to your manuscript, taking into account all the changes suggested by the reviewers. After a careful review by myself, I do not consider it necessary to re-invite reviewers, since all comments have been taken by authors, I am satisfied with this version.